# What influences cancer treatment service access in Ghana? A critical interpretive synthesis

Chloe Zabrina Tuck [ID],[1] Robert Akparibo,[1] Laura A Gray [ID],[1] Richmond Nii Okai Aryeetey,[2] Richard Cooper [ID][1]

¹School of Health and Related Research (ScHARR), The University of Sheffield, UK
²Population Family and Reproductive Health, University of Ghana, Legon, Greater Accra, Ghana

**Correspondence to**
Dr Chloe Zabrina Tuck;
cztuck1@sheffield.ac.uk

## ABSTRACT

**Objectives** Multiple social-cultural and contextual factors influence access to and acceptance of cancer treatment in Ghana. The aim of this research was to assess existing literature on how these factors interplay and could be susceptible to local and national policy changes.

**Design** This study uses a critical interpretive synthesis approach to review qualitative and quantitative evidence about access to adult cancer treatment services in Ghana, applying the socioecological model and candidacy framework.

**Results** Our findings highlighted barriers to accessing cancer services within each level of the socioecological model (intrapersonal, interpersonal community, organisational and policy levels), which are dynamic and interacting, for example, community level factors influenced individual perceptions and how they managed financial barriers. Evidence was lacking in relation to determinants of treatment non-acceptance across all cancers and in the most vulnerable societal groups due to methodological limitations.

**Conclusions** Future policy should prioritise multilevel approaches, for example, improving the quality and affordability of medical care while also providing collaboration with traditional and complementary care systems to refer patients. Research should seek to overcome methodological limitations to understand the determinants of accessing treatment in the most vulnerable populations.

## INTRODUCTION

Cancer is a growing burden in low and middle income countries (LMICs).[1 2] Despite efforts by the WHO to prioritise tackling cancer inequity, hurdles remain due to the limited evidence to inform cost-effective decision-making and the high expense of cancer control.[1] In Ghana, cancer treatment is focused in large referral centres in major cities, with disparity in resources and health worker expertise in rural areas and limited coverage of the National Health Insurance Scheme (NHIS).[3–7] Policy efforts to expand cancer services are further hindered when patients prioritise traditional alternatives over orthodox cancer services.[8] Multiple social-cultural, economic and health system factors can influence how patients access, navigate and choose suitable cancer care in Ghana.[3] In addressing this, there is a relative lack of public health surveillance data. There have been recent attempts to reconcile this,[9] but a comprehensive understanding remains elusive. One alternative approach is to consider the relevance of theory.

### Socioeconomic model and candidacy framework perspectives on cancer treatment access

One important way of understanding the complexity and various factors previously described in relation to cancer treatment service access is to consider this in terms of the socioecological model.[10] This has been used in many settings to map barriers to healthcare engagement from a systems perspective.[11 12] The socioecological model considers the individual within an ecosystem of intrapersonal, interpersonal, community, health organisational and policy influences. This has been applied extensively to map systems factors since it was developed by Bronfenbrenner,[10] including health behaviours in several African settings.[11 12] The process of an individual accessing cancer services is dynamic and delays in access can occur at multiple

stages. Patients may experience barriers presenting at services, negotiating the care pathway, being offered and accepting treatment. A critical interpretive synthesis (CIS) exploring health access in the UK[13] highlighted that this involves dynamic interactions between the individual, the health environment, and health professionals. The term 'access' often overlooks this dynamism, while terms such as 'uptake' provide a narrow view that overlooks patient demand and service navigation afterwards.

⇒ A strength of this study is the combination of purposive and systematic searches, and the reflexive approach to developing the search strategy which enabled it to cover a wide range of articles.
⇒ Additionally, the critical interpretive synthesis involved a critique of the literature to identify methodological limitations and research gaps.
⇒ However, as only published academic articles are included in this study, it may overlook other forms of evidence, including locally generated and day-to-day working understandings.
⇒ The interpretation of the evidence will reflect the inherent biases in world view of the lead author.

The candidacy framework provides a multistage interacting process of patient access holistically. For example, negotiation of services is overlooked in asylum seekers and refugees.[14] van der Boor and colleagues described candidacy in two broad stages: 'access' (identification of eligibility, navigation of services) and 'negotiation' (permeation of services, appearance, adjudications, offers and resistance, and dynamic interactions with local services).[14] Candidacy has been applied to understand patient-doctor interactions influencing cancer health seeking behaviours[15] and in an African context.[16]

This paper builds on the issue of factors relating to cancer treatment service access in LMIC settings such as Ghana by presenting the findings of an evidence review that was informed by theory. The primary aim was to systematically review and critique literature from a systems perspective to understand factors influencing cancer treatment service access in Ghana. A further aim was to assess the strengths and limitations of methods associated with existing research relating to this topic.

## METHODS

Using the RETREAT (Review question, Epistemology, Timescale, Resources, Expertise, Audience/purpose, Type of data) framework,[17] CIS was considered to be the most suitable approach.[13] CIS has been used in a variety of policy and health service settings[18–20] and combines systematic and purposive approaches to identify multiple types of evidence and identify themes following an evidence critique. This involves considering how the problem has been constructed, underlying assumptions and epistemology, and how this has influenced the methodology and conclusions.[13]

### Search strategy and literature search

The search strategy to find articles on access of adult cancer treatment in Ghana was developed using the question framework PerSPECTiF (Perspective, Setting, Phenomenon of interest, Environment (optional Comparison) Timing, Findings)[21] in consultation with an information specialist. First, primary systematic searches were undertaken. This was tested and refined through pilot searches, before conducting comprehensive searches in Medline (via OVID), Web of Science, CINAHL and African Index Medicus. These databases were chosen following University of Sheffield librarian advice, and after the initial database scoping exercise in Medline (via OVID) and Google Scholar. The database search strategy and terms used can

be found in online supplemental table 1. Initial searches were conducted on 26 March 2021 and the databases were searched for updates on 29 March 2022. Search terms were composed of multiple equivalent thesaurus terms and phrases to cover three elements: Ghana, health service access/uptake of services and cancer. Hand searches were performed using citation follow-up, identified relevant individual journals and in the reference listed of included papers.

### Study selection

The lead author (CZT) screened all titles and abstracts using an agreed inclusion criteria, while two other authors (RA, RC) conducted quality checks of 10% of the sample screened. Any disagreement was discussed and settled among authors. Initial screening highlighted ambiguity in the screening criteria, which was further refined to ensure consistency prior to formal selection. Inclusion criteria included only primary research conducted with a 10-year time frame to align with Ghana's increased policy interest in efficiently expand national health insurance packages.[6] Initial screening highlighted the need for a focused exclusion criteria which was again informed by the question framework PerSPECTiF.[21] Applying the PerSPECTiF framework, the phenomenon of interest ('access') was defined holistically through the candidacy framework.[13] Thus, article screening sought to include articles relating to access throughout the entire patient pathway. The setting included all levels of the socioecological model to provide a systems perspective. Potentially qualifying abstracts were read in full, and only the full texts papers that meet the review inclusion criteria were included and reviewed.

### Data extraction and synthesis

Data were extracted from included papers to facilitate decision-making and an audit trail (see data extraction in the online supplemental material). The lead author (CZT) used a standardised data collection form to extract data from the included studies. To eliminate data extraction bias, two reviewers (RA and RC) checked 10% of the extraction. There was no discrepancy observed between the lead author extraction and the sample reviewed. Key data extracted were setting, approach, population and sample, methods design, sampling, data analysis, cancer and stage studied, and the corresponding texts cross-tabulated against the socioecological model[10] and the candidacy framework. Data were collected as line of arguments.[13] First order constructs (taken directly from the data in articles) and second order constructs (author reports from articles) were extracted and separately noted within the framework. It was noted where study authors made further (secondary) inferences and assumptions from data that were not primary findings, but relevant to the themes. Researchers' limitations were recorded. The data were segregated into qualitative, quantitative and mixed methods studies and each interpreted qualitatively. A synthesising argument[13] was

applied to map first and second order findings across studies to interpret the evidence and to create new concepts that draw on the whole body of evidence (third order/synthetic constructs). Inferences were mapped across the studies to enable the body of evidence to be critiqued by research question construction, methods, and conclusions and how these fit into the general findings, to identify trends in the literature and limitations with current approaches. To bring together themes in candidacy, the candidacy framework was summarised into three main stages. van der Boor and White used two stages[14]; however, as themes were identified, it was noted that treatment acceptance and the interactions around this over time play a pertinent role in the patient pathway in Ghana. Thus, the adapted three-stage model also notes the importance of the dynamics of treatment acceptance. As part of the synthesis process, the primary literatures from the data extraction were revisited and reinterpreted with emerging evidence to ensure critical details or limitations were not missed.

## Critical appraisal

A streamline critical appraisal for major and methodological flaws was conducted using the critical appraisal checklist published in Dixon-Woods *et al*.[13] The quality of quantitative and qualitative findings was assessed in terms of reliability and trustworthiness.[22] In line with the

CIS, articles were not favoured based on quality alone but contribution of rich insights. The CIS deals with weak evidence through including a critique of methods and approach. Lead author (CZT) appraised all the included papers with 10% of the sample being cross-checked by RC.

## Patient and public involvement statement

No members of the public or patients were involved in this research.

## RESULTS

### Search results

Systematic searches in four databases and in six journals performed in March 2021 (updated in March 2022) identified 312 citation. After duplicate removal, 203 potentially relevant abstracts were screened, subsequently 78 articles were identified for full text screening. A further 16 abstracts were identified for full text screening following citation and reference searching. Twenty-eight articles were selected for inclusion (see PRISMA (Preferred Reporting Items for Systematic Reviews and Meta-Analyses) flow diagram in figure 1). These comprised 15 qualitative, 12 quantitative studies and 1 mixed methods study.

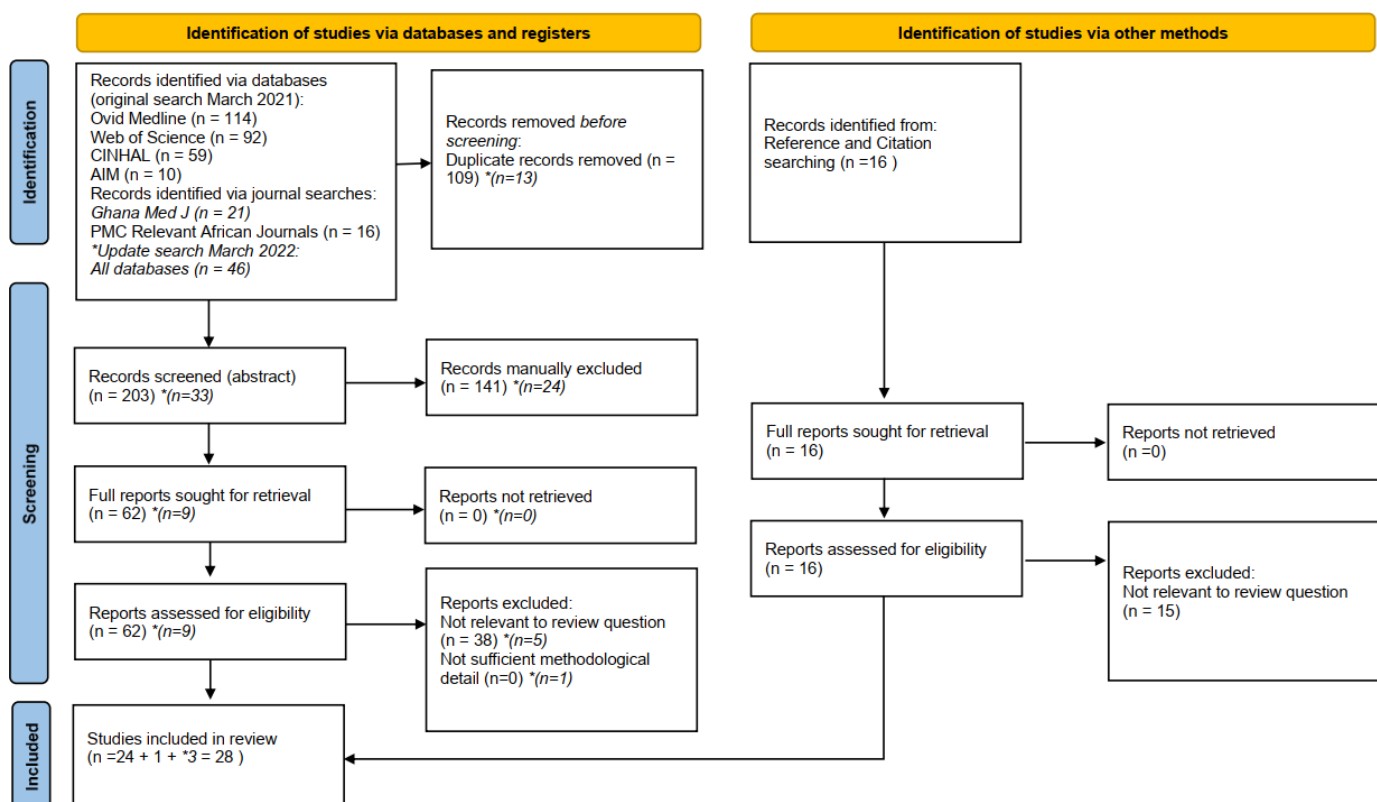

**Figure 1** PRISMA (Preferred Reporting Items for Systematic Reviews and Meta-Analyses).*Italicised numbers are from search update conducted March 2022 to identify newly published literature. Diagram adapted from from: Page MJ, McKenzie JE, Bossuyt PM, Boutron I, Hoffmann TC, Mulrow CD, *et al*. The PRISMA 2020 statement: an updated guideline for reporting systematic reviews. BMJ 2021;372:n71. doi: 10.1136/bmj.n71. For more information, visit the website (http://www.prisma-statement.org/).

A summary of the articles reviewed is included in the online supplemental table 2. Applying the candidacy framework and socioecological model, key themes were identified and are presented below. The evidence mapping table (online supplemental table 3) highlights the candidacy stage and level within the socioecological model that the articles addressed, as judged by the lead author.

### Accessibility defined through a 'candidacy' lens

The candidacy framework[13] proved valuable in assessing how healthcare access has been approached, from a holistic perspective. Treatment acceptance was a key emerging issue where there was a gap in understanding. Within treatment acceptance there were multiple aspects—delays accepting, interruptions, choosing alternatives, and incompletion or loss to follow-up, noncompliance, refusal for referral and non-acceptance of diagnosis. This was a dynamic process.

Although the full candidacy process has been considered in research, there were key gaps in how it had been approached and some aspects warranted further exploration. Seven studies aimed to explore delays with initial contact with cancer services on identification of need (presentation), yet the reasons for eventual nonpresentation could not be explored in most studies that were clinic based, as all patients surveyed eventually reached treatment centres. Two studies quantified acceptance as having complete follow-up and treatment completion. A further four noted delays and high nonacceptance/loss to follow-up but did not formally explore them at an individual psychosocial level as this was not within the research aims. Twelve qualitative studies explored individual barriers to accepting care. However, this was not always the primary focus but emerged in the findings.[23 24] As these were sampled from a clinic, they represented patients who eventually (despite delays or interruptions) accessed treatment, so the enablers and barriers in those who ultimately dropped out was unknown. Therefore, in-depth qualitative psychosocial information on treatment incompletion was not collected. Although the literature on acceptance was predominantly breast cancer related, there was some limited evidence it occurred in other cancers, but the extent and reasons for this were not explored.

### Barriers and enablers of cancer service access interpreted through the socioecological model

The findings on enablers and barriers to candidacy for cancer treatment were mapped using the socioecological model to consider the Ghanaian health ecosystem. These are summarised in figure 2.

### Intrapersonal
#### Financial barriers

Inability to afford treatment was reported as a barrier leading to delays in and non-acceptance of care. This was also noted by traditional herbalists[25] and health workers.[3]

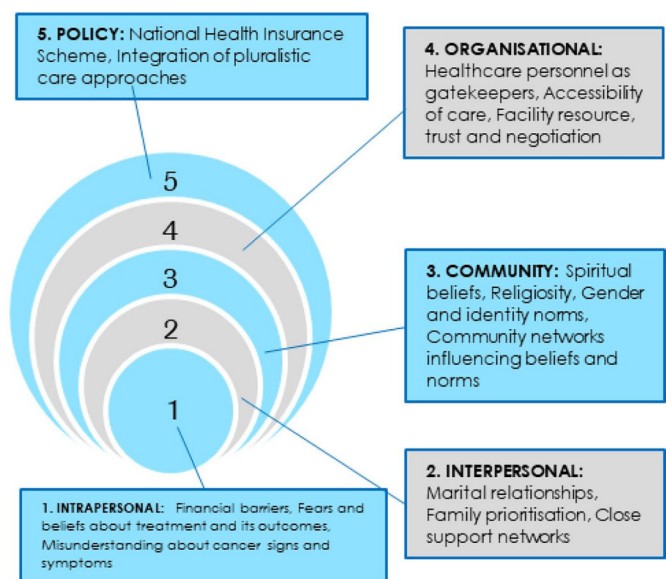

**Figure 2** Key influences on candidacy for cancer treatment mapped against the socioecological model.[10–12]

Yamoah *et al*[26] found that it encompassed socioeconomic factors, travel costs and lost work. Although two studies indicated it was a greater barrier in those from a low income,[27 28] Sanuade *et al*[29] suggested this was regardless of socioeconomic status. Four studies[23 28–30] demonstrated that high cost of medical treatment led to use of herbal and traditional alternatives. Prioritisation of finances on family led to delays in accepting treatment.[23 24 31] Financial barriers impacted negotiation of care[31]; those from a lower income were more likely to experience longer wait times.[32]

### Fears and beliefs about treatment and its outcomes

Obrist *et al*[33] found patients who believed in the efficacy of treatment were more likely to complete treatment in single variable analysis, although this was not significant in multivariate models where potential confounding factors were controlled for. Fears and beliefs pose individual level barriers from qualitative finding. This included fear of treatment, medicines and the outcomes.[8 24 28 29 34] For example, for breast cancer, loosing breasts, womanhood, female identity.[24 28 29] Fears around institutional trauma suggested lack of trust health facilities.[29] Healthcare professions deliberately miscommunicated to avoid patient fear and drop out.[35]

### Misunderstanding about cancer signs and symptoms

Lack of knowledge about cancer signs and symptoms led to delays in seeking medical treatment.[8 24 28 31 36–38] With breast cancer, lumps were not regarded as serious when painless and sometimes considered part of normal tissue.[8 24 28 31] This was influenced by beliefs held in the community and within patients' social networks.[8 31 36] Lay beliefs were influenced by the terminology used for breast cancer in the local dialect, which led to poor understanding.[39]

## Social demographics associations

Evidence was conflicting as to whether age, religion and ethnicity impacted stage of diagnosis, wait times and treatment completion, which may associate differently in different cancers given different demographics and natural histories.[26 32 33 40–43] As ethnicities often cluster predominantly in different regions, the potential confounding of local health system and environment should be checked in future studies. Some evidence indicates that lower education is associated with the presenting of larger tumour masses[43 44] and waiting longer for treatment,[32] but this was not a consistent influencer, and whether it was associated with treatment acceptance was not explored. One study exploring treatment pathways for young people with chronic diseases suggested community beliefs were more influential than educational status.[36] Another found that community beliefs and norms influenced the perceptions of breast cancer regardless of socioeconomic status.[39] Despite clear financial barriers to treatment, there was no evidence that income status and occupation were associated with presentation and acceptance of treatment, but low income status was associated with increased wait times in one study.[32]

## Interpersonal

### Marital relationships influence treatment seeking

Qualitative evidence in female cancer suggests husbands influence their wives' treatment seeking behaviours and acceptance,[8 24 27 29 31 38 39 45] by controlling financial decisions about treatment.

### Family prioritisation delayed treatment

Women prioritised other activities linked to their economic, family and social roles, such as working for more money, treating children and paying school fees. These lead to delays presenting for, negotiating and accepting treatment.[8 23 24 28 31] Caring for others meant patients put the needs of others first, neglecting their own health. This aligns with findings on the unaffordability of treatment.

### Close support networks influence treatment access

Patients understanding about cancer, its causes and how they engage with care was influenced by close friends and family.[8 23 28 34 36–38 45] Misinformation could lead to late presentation, delay help seeking and use of alternatives.[8 23 36 37] For women, lack of husband, family and friend support delayed treatment seeking.[27 45] Familial financial support was an enabler for some to seek treatment,[23 45] whereas family neglect may impeded access.[3]

## Institutional

### Healthcare personnel as gatekeepers to medical and alternative care

Poor detection at primary health facilities, community pharmacy and private settings may have delayed diagnosis.[3 8 23 31 36 44–46] Some women sought over the counter medications for pain management.[37] Seeking assistance from someone other than a nurse or doctor was associated with a larger mass at diagnosis for breast cancer, which could include a diverse mix of pluralist and community supports.[44] Some health professionals also advised herbal alternatives, delaying medical cancer treatment.[29] A mixed method study found an inability for facilities to diagnose cancer, improper documentation and filing of patient folders and workload—likely exacerbated by a shortage of healthcare workers trained in oncology outside of major tertiary centres.[3] In agreement, qualitative studies with patients found misdiagnosis were common.[37 38] There were delays due to the complex referral process, waiting a long time to get results, having to go to many hospitals and laboratories to be diagnosed, and consultant rescheduling.[37 38] Delays between referral and starting radiotherapy were suggested to be due to resource availability, while irregular medicine supply also meant patients had to source medicine outside hospitals at high cost.[23 47] Patients showed negative perceptions of the care system and professionals.[27 29 29 45] Patients perceived treatment delays due to workforce shortage, hospital machines breaking and medicines shortages.[29] These beliefs appear to contribute to a lack of confidence and trust in the health system. Non-completing patients were more likely to harbour negative views such as that the unavailability of cancer medicines delayed their treatment.[33] Doubts in the efficacy and disappointment with conventional treatment created barriers to seeking treatment[27] and influenced use of pluralistic treatments.[30] Fear of radiation led some not to receive clinically recommended treatment.[30]

## Community

The body of literature showed the strong role community beliefs and norms played in shaping access to cancer care. These were interconnected with personal perceptions and health system factors.

### Spiritual and traditional beliefs about cancer causes and treatment

In an overwhelming majority of the literature, patients assigned their cancer diagnosis to spiritual causes, which led patients to seek traditional herbal and spiritual treatments, delaying presentation and interrupting medical treatment at multiple stages. This was associated with financial barriers to conventional treatment,[23 28 29] advice from supports such as spouse,[45] health workers,[29] religious messages,[8 29] community networks and beliefs.[8 29 31 34 36 39] Additionally, alternative therapies were often perceived as more available and acceptable, seen as efficacious.[27 28 39 47]

### Religiosity plays a diachronous role

Religious beliefs, messages and leaders influenced alternative therapy use[8 29] and caused delays.[8 29 37] Yet, religious leaders were identified within patients' trusted support networks[8] and their advice facilitated medical presentation.[8 24] Some studies found the church played a supportive role, encouraging women to present at services and providing financial assistance to low-income families.[24]

## Gender and identity norms

For women with breast cancer, mastectomy was associated with a fear of 'diminished sexuality and femininity' which led many women to delay treatment after seeing an oncologist.[24 28 29 31] Unaddressed fears about fertility loss may have increased dropout.[35] However, the barriers around identity may differ in other tumours and population groups.

## Community networks influenced beliefs and norms

Common misconceptions, beliefs and behaviours held by patients were reinforced by community and social network beliefs.[8 29 31 36] Interlinked with community beliefs about cancer is self and socially experienced stigma due to the cause of the disease being spiritual: a curse, misendeavour or the patient being a witch.[24 27 39] This led to patients seeking traditional herbal and spiritual treatments, while creating shame and secrecy. A retrospective survey found patients who did not complete treatment were more likely to answer they do not know if they were fearful of their community response. However, this likely reflects uncertainty by next of kin respondents, who were substantially higher in 'non-completing treatment' groups.[33]

## Policy

### NHIS inclusion of cancer care

Based on the financial barriers to treatment reported by patients and healthcare workers, lack of cancer care inclusion within the NHIS was inferred to lead to treatment refusal and delays.[3] This was the case for cancers not covered by the scheme,[26 42 48] as well as breast[23 24] and cervical cancer.[27 40] Patients with breast cancer were unanimously frustrated that the NHIS did not cover substantial amounts of treatment and discussed the huge financial burden, especially of chemotherapy drugs.[23 24] This meant some women could not start treatment on time.[34] This was aggravated by medicines stock-outs,[3 23 29 47] requiring purchase elsewhere at additional cost.[23 47] Not being insured was significantly associated with a shorter wait time for breast cancer treatment,[32] which could reflect preferential treatment to those paying upfront due to delays in the administrative process of reimburse NHIS funding. Knowledge of hormone receptor status predicted complete treatment follow-up, as this service is offered at a cost.[41] Healthcare professionals acknowledged costs were barriers for patients but struggled to broach such topics.[35]

### Integration of pluralistic care approaches

Given the prominent roles of traditional, herbal and spiritual care improved integration with the orthodox health system could improve patient access.[25 28]

Thirty-eight per cent of clinical workers surveyed in Ghana attribute treatment disruptions to traditional medicines use.[3] Reported use of traditional healers was a significant predictor of late presentation after other variables were controlled for.[44] An assessment of factors associated with treatment completion found visiting a traditional healer was a significant predictor of not completing treatment.[33] Although alternative support could be concurrent to orthodox medical intervention, over 50% of complementary and complementary medicine practitioners surveyed indicated they did not let patients seek other care alongside[25] and 63% of customers said they declined orthodox therapy while using such therapies.[30] Nevertheless, one study found that although 12.2% seek alternative therapies, this only partially explains high rates (73.1%) of loss to follow-up.[41] While traditional herbalists are considered health professionals, with some services integrated into the Ghanaian health system, poor knowledge of cancer causes and symptoms, and treatment and reluctance to refer to other services are barriers to providing their patients with timely appropriate care.[25] This is influenced by a perceived reluctance collaborate from other health system components.[25] Mburu *et al* suggested the interaction between treatment approaches is non-linear as acts at multiple pathway stages.[37]

## Critique of the evidence

As part of the CIS approach, a critique of the literature was conducted to identify themes in methods, assumptions, theories and analysis to identify methodological limitations and research gaps for further studies. A summary of the studies characteristics is displayed in figure 3.

This review accurately represents evidence in Ghana, which has a high breast cancer contribution. Most studies focus either on presentation delays or treatment interruptions, barriers and treatment delays. Although many studies note treatment incompletion and loss of follow-up, only two assess this directly[33 41] and they note challenges in data collection, there are no qualitative studies exploring definitive incompletion. Eighty-eight per cent (25/28) of studies were based in tertiary treatment clinics in the Greater Accra and Ashanti region, so may not reflect those in other regions. As 89% of qualitative studies sampled purposively from tertiary clinics, this led to biased sampling. They omit those who did not attend, those who dropped out without contact or cannot reach treatment centres. Sampling patients in clinics represents those who eventually presented for and accepted treatment. Barriers could be experienced differently in the under-represented population. As some studies had a small sample and were predominantly Christian, there was limited ethnographic diversity, which may influence generalisability. Findings on the social demographic traits linked to barriers accessing treatment were inconclusive.

Hypothetical inquiry was found to be common across studies. This may lead to error if an individual is not able to accurately predict their behaviour in an unknown situation,[49] which may be the case as cancer is stigmatised and not talked about openly.[50] Further work should explore and seek to understand the impact cognitive bias may have when using hypothetical situations. The gaps in research identified are summarised in box 1.

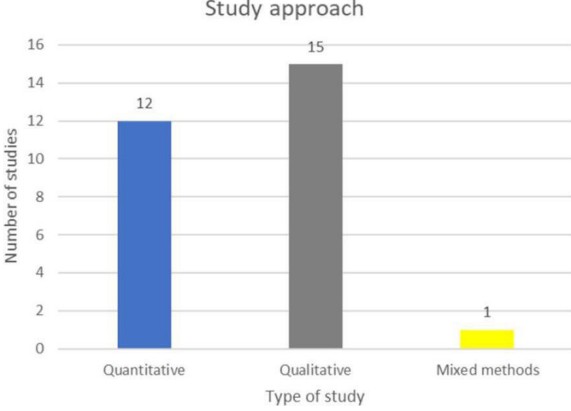

Study approach

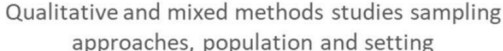

Qualitative and mixed methods studies sampling
approaches, population and setting

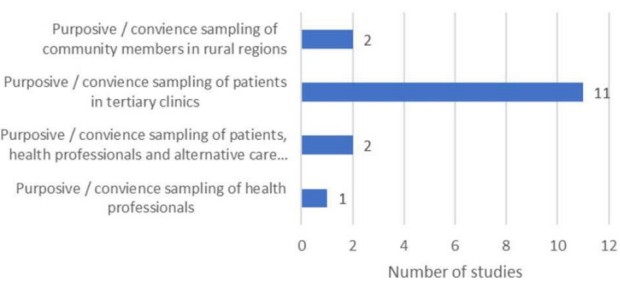

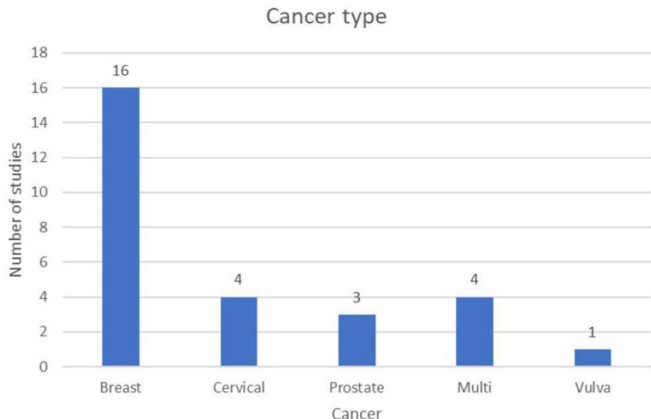

Cancer type

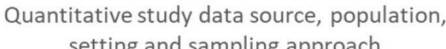

Quantitative study data source, population,
setting and sampling approach

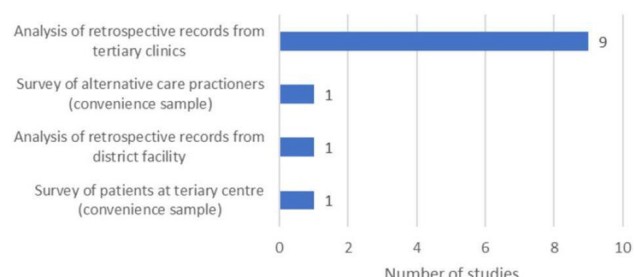

**Figure 3** Summary of characteristics of included studies.

## DISCUSSION

This study used CIS to review multiple types of qualitative and quantitative evidence from literature on access to cancer treatment services in Ghana. Applying 'candidacy' enabled the dynamic and continuous process of accessing, negotiating and accepting treatment to be explored within the Ghanaian social, economic and policy environment. It highlighted determinants of cancer treatment service access in Ghana are interlinked and within each stage of the socioecological model. There is a research gap in understanding the determinants of accessing treatment in the most vulnerable populations due to methodological limitations.

> **Box 1    Themes and research gaps identified through critiquing the evidence**
>
> ⇒ Most studies are situated at tertiary clinics, so may not represent rural regions.
> ⇒ Sampling from tertiary clinics means populations who do not present, negotiate the referral pathway from local services and eventually choose treatment are not represented. Understanding what influences access in these missing populations warrants further study.
> ⇒ Treatment dropout is frequently observed but the reasons for this are not sufficiently explored.
> ⇒ As most studies focus on breast cancer, there is need to understand the extent of treatment drop out across all cancers and which factors influence this.

Through this approach, we were able to critique the literature, highlighting trends in methodology and gaps in evidence for future study. The CIS enabled detailed context-specific insights as well as identifying limitations in research approaches, data collection and acquisition challenges to inform future research. A reflective and iterative approach to broaden the breadth of evidence, interpretation and assimilation, was taken. This was particularly valuable for integrity in research in West African, due to epistemic injustice in how knowledge is perceived and interpreted.[51] This is the first study to have explored the applicability of the candidacy framework of healthcare access to a Ghanaian setting. Although this has proven valuable in other African settings,[16] this model was developed in a UK setting, therefore it was important to acknowledge pre-existing bias in perspective and thus to critically assess this framing and how it might impact the interpretation of results.[51] Selecting globalised frameworks (over those locally synthesised) can lead to interpretive marginalisation. Another limitation of the framework was that we found differences in conceptualising and describing patient engagement pathway with cancer services between studies meant that ascertaining candidacy categories for each required researcher interpretation.

Only published academic articles were included in this study, which may overlook other forms of evidence, including locally generated and day-to-day working understandings. However, this was minimised through

an iterative approach to include multiple databases, targeting local journals and supplementary searches.

A methodological challenge was the vague and broad nature of terms relating to 'accessing' healthcare. Thus, there was a need to balance breadth of search with pragmatism to formulate a multistep search strategy. However, this may mean not all relevant literature was uncovered. Additionally, the critical interpretive nature of the review meant the evidence interpretation, conducted by the lead author, will reflect their inherent biases in world view.[22] This review was focused on access to cancer treatment in Ghana exclusively, so it is uncertain whether the findings translate to other contexts.

The evidence highlighted financial barriers to cancer treatment access, which interacted with cultural factors and societal influences, such as norms around managing household finances, prioritisation and cultural acceptance of alternative medicines. Globally, catastrophic costs (defined as greater than 40% household effective income) due to non-communicable disease health expenditure are prevalent in LMICs, leading to individuals not taking medicines and impoverishment.[52] This exacerbates inequities, having the greatest impact on the poor and leads to detrimental coping strategies.[53]

Poor patient clinician relationships have been found to lead people to seek traditional medicines alternatives in Ghana.[54] Traditional medicine practitioners were seen to offer more patient-centric, holistic care which was more comforting. A lack of trust was noted for orthodox facilities, which may reflect cultural beliefs as well as past healthcare experiences. An evidence synthesis across LMICs found modern medicines viewed to be harmful and ineffective; suspicion and mistrust of biomedicine lengthened delay and led to alternative use, and the impact of this may be exacerbated in the most vulnerable.[50] A qualitative exploration of influences on traditional medicines use in Ghana found their 'pull' by accessibility and alignment with cultural beliefs, whereas scepticism of biomedicine may push people from orthodox healthcare.[54]

At a community level, spiritual beliefs about the origin of cancer interacted with personal perceptions at the individual level. Notions and beliefs typically held in the community can be ascribed as lay explanatory models of disease. In accordance with Kleinman,[55] lay models of disease can differ from biomedical models based on social experiences and impact how individuals interpret and act on their condition. Community factors influenced explanatory models for hypertension in rural northern Ghana and impact treatment access.[56] Similarly, in this review, explanatory models created stigma leading to secrecy and selection of traditional medicines over biomedical intervention.

At a policy level, two key themes where reforms could improve cancer treatment uptake stood out: (1) greater inclusion of cancer treatments within the NHIS, (2) enhanced integration of traditional medicines to provide complementary options to medical care for cancer.

Despite the NHIS aim of achieving universal health coverage for all, there has been notable disparity, with the lowest coverage concentrated in the poor.[57 58] Multi-dimensional barriers due to poverty, dissatisfaction and distrust of the health service and staff may prohibit enrolment.[59] Furthermore, catastrophic spending due to out of pocket costs remains high.[60] The NHIS plan seeks to cover a considerable amount of the local disease burden and since its nascence the inclusion list has been revised in response to transitions in disease burden and advancing treatments.[61 62] Despite breast and cervical cancer in theory being covered,[63] it is widely noted women still face considerable financial burden, which this study further highlights. The burden in men however remains less clearly mapped. The high expense of cancer medicines poses a challenge to decision-makers, who must weight costs and benefits when deciding on how to invest health budgets. Approaches such as the health technology assessment platform recently established in Ghana could help prioritise high-cost medicines.[64 65]

Another policy area identified in this review was coordination between medical services and traditional medicines. Studies in Ghana have shown traditional medicine users are more likely to be poor and not insured on the NHIS.[66] The NHIS currently provides services through a plural system of public, faith-based, governmental and private facilities, and includes some traditional medicines.[57] Although lack of harmonisation of traditional medicines with the healthcare system[67] was reported in this study, there have been multiple reforms to this end since the Traditional and Alternative Medicine Directorate was established under the Ghanaian Ministry of Health in 2001.[68] Still coordination is hindered by a lack of professional respect from other health professionals.[67] Co-current use of alternative forms of care has been found in pregnant women in Ghana. It was highlighted the individual psychosocial and emotional support they provide, which this study found can be lacking patient interactions with orthodox medicine, despite being key for candidacy in cancer treatment.[15 69]

The African Union has calls to recognise the importance of traditional medicines and a Global Centre for Traditional Medicines has been established in India.[70 71] However, there remain challenges in how traditional and complementary medicines are perceived and deemed efficacious. Barry[72] suggests differently constructed modes of evidence are needed for traditional medicines, as scientific evidence through clinical trials offers a reductionist, narrowly defined view of evidence, that overlooks the role of lived-in social experiences, advocating for the 'expanded epistemology of science'.[73] Future policies should seek to improve the affordability and quality of publicly provided medical care while harmonising with complementary treatments that align with community beliefs.

Future research is needed to address the research gaps identified. First, to understand the extent of treatment non-adherence across all cancers and what individual,

social and health system factors impact this. Second, there are methodological limitations in understanding the views of those who do not attend clinics, which may represent the most vulnerable. Researchers should seek approaches to overcome this which are suitable within the local context.

**Acknowledgements** The authors would like to thank the School of Health and Related Research Library team for their expert advice in designing the search protocol, as well as all researcher staff and graduate at the Universities of Sheffield and Ghana who commented or provided feedback on preliminary results.

**Contributors** CZT, RA, LAG and RC conceptualised the study. CZT led in designing the study, conducted literature search to collect data, performed the data analysis and wrote the first draft. RA, LAG and RC contributed to selecting literature for inclusion in the study and advised on data collection and analysis. RC, RA, RNOA and LAG reviewed the initial and final draft manuscript and made intellectual inputs to improve quality. All authors read and approved the final manuscript. CZT is responsible for the overall content as guarantor and accepts full responsibility for the work and controlled the decision to publish. The corresponding author (CZT) attests that all listed authors meet authorship criteria and that no others meeting the criteria have been omitted.

**Funding** This research was funded by Wellcome as part of a PhD studentship (108903/B/15/Z).

**Competing interests** None declared.

**Patient and public involvement** Patients and/or the public were not involved in the design, or conduct, or reporting or dissemination plans of this research.

**Patient consent for publication** Not applicable.

**Ethics approval** Not applicable.

**Provenance and peer review** Not commissioned; externally peer reviewed.

**Data availability statement** Data sharing not applicable as no datasets generated and/or analysed for this study. Data sharing not applicable as this is a review of literature, and no datasets were generated.

**ORCID iDs**
Chloe Zabrina Tuck http://orcid.org/0000-0003-3525-6295
Laura A Gray http://orcid.org/0000-0001-6365-7710
Richard Cooper http://orcid.org/0000-0001-5110-0384

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
