## [Reviewer comments · BMJ Open]

ARTICLE DETAILS

TITLE (PROVISIONAL)	What influences cancer treatment service access in Ghana? A critical interpretive synthesis
AUTHORS	Tuck, Chloe; Akparibo, Robert; Gray, Laura; Aryeetey, Richmond; Cooper, Richard

VERSION 1 – REVIEW

REVIEWER	Agyemang, Linda University of Southampton, Health Sciences
REVIEW RETURNED	16-Jun-2022

GENERAL COMMENTS	General comments Thank you for providing an extensive and critical exploration of the current evidence on the socio-cultural factors impacting access to cancer treatment in Ghana. Nonetheless, I have some concerns which I have highlighted below: In the abstract section, perhaps you could be specific about the socio-ecological determinants which are interacting and whether they do so to deepen access barriers. Again, in the conclusion part of the abstract, you could be specific about those multi-level approaches that might be appropriate for improving affordability of medical care. Methods While the search terms were mentioned, no evidence of search strategy was provided. There was no mention of whether the review was registered or not. Although it was mentioned papers within a 10-year time frame were reviewed, it was not clear when each database was last searched. With regards to access, Candidacy has been described as 'identification of eligibility' and 'navigation of services' but it is unclear how the search terms reveal the 'holistic definition of access'. Good mention of selected databases but there was no mention of why they were chosen. Perhaps you might want to consider proving the full meaning of RETREAT and PerSPECTiF In the data synthesis, it was unclear what data on study characteristics were extracted. While a thorough critique was provided for the evidence synthesis, not all the included studies had their risk of bias presented. Results In the section on 'accessibility defined through a candidacy lens', perhaps you might want to include study references. Community The role of the community in shaping access to care is presented accordingly but it is interesting that there was no mention of the role of the media (e.g., television and radio) in influencing treatment access. The role of the media in accessing medical care for breast cancer was a finding in Agbeko et al. (2020).
---

	Specific comments Introduction Line 40-41 You may consider removing these two sentences as they seem superfluous because lines 35-36 have already highlighted how 'access' might obscure the dynamism between individuals, their health environment and healthcare professionals. Results (Page 5) Line 46 Perhaps you might want to be more specific about the number of qualitative studies. Line 48 You might want to consider rewriting this. For example, 'In ... studies (insert reference), this was not the primary focus but emerged in the findings. Institutional (Page 7) Line 30 You might want to consider re-writing OTC in full. Discussion (Page 11) Line 6 Please correct the typo ' A lack of trusted...
--	---

REVIEWER	Akuoko , Cynthia Christian Service University College, Nursing and Midwifery
REVIEW RETURNED	19-Jul-2022

GENERAL COMMENTS	Thanks for the opportunity to review this study. Neatly written paper. Some few concerns Abstract - Please be consistent with the use of the word Social-ecological or socio-ecological. The paper will benefit from proofreading to correct the grammatical and semiotical errors. Titles for some of the figures were missing.
--

VERSION 1 – AUTHOR RESPONSE

Comments	Response	Text	Page reference
Reviewer: 1			
Thank you for providing an extensive and critical exploration of the current evidence on the socio-cultural factors impacting access to cancer treatment in	Thank you for your feedback, this is hugely valuable in further shaping our manuscript.		

Ghana. Nonetheless, I have some concerns which I have highlighted below:			
In the abstract section, perhaps you could be specific about the socio-ecological determinants which are interacting and whether they do so to deepen access barriers.	Thank you, we have clarified this in the abstract. It is also mentioned in the results, and we have further clarified in the discussion to avoid ambiguity. We do not have evidence on whether this deepened barriers, which may also be subjective and individualised.	Community level factors influenced individual perceptions and how they managed financial barriers (such as seeking traditional medicines). At a community level, spiritual beliefs about the origin of cancer interacted with personal perceptions at the individual level.	P2 P11
Again, in the conclusion part of the abstract, you could be specific about those multi-level approaches that might be appropriate for improving affordability of medical care.	We have clarified what we mean by multi-level approach, however we have not been more specific as for the interventions as it was not in the remit of the study	Future policy should prioritise multi-level approaches, for example improving the quality and affordability of medical care while also providing collaboration with traditional and complementary care systems to refer patients	P2
While the search terms were mentioned, no evidence of search strategy was provided.	We have added a systematic search strategy and search terms in the supplementary material	The database search strategy and terms used can be found in Supplementary Material.	P4 Supplementary material page 1
There was no mention of whether the review was registered or not.	The review was not registered as it was not deemed appropriate for a CIS, where the researched phenomenon is seen as evolving through a reflexive approach instead of adhering to a predetermined protocol. This is considered acceptable practices for CIS, scoping and mapping reviews (see Booth et al		

	2016 in their book titled systematic approaches to a successful literature review 2 nd edition).		
Although it was mentioned papers within a 10-year time frame were reviewed, it was not clear when each database was last searched.	This was included in the results, however we agree it would be helpful to state this also in the methods section. Therefore this has now been added.	Initial searches were conducted on 26/03/2021 and the databases were searched for updates on 29/03/2022.	
With regards to access, Candidacy has been described as 'identification of eligibility' and 'navigation of services' but it is unclear how the search terms reveal the 'holistic definition of access'.	Thank you for noting this. We have included the search terms in the supplementary material. Given the limitations balancing specificity and comprehensively in the search terms, the screening process was key to ensuring a holistic definition of access was maintained. We have indicated this in the methods.	Apply the framework developed using PerSPECTiF, the phenomenon of interest 'access' was defined holistically through the candidacy framework (13). Thus, article screening sought to include articles relating to access throughout the entire patient pathway. The setting included all levels of the socio-ecological model to provide a systems perspective.	P4
Good mention of selected databases but there was no mention of why they were chosen.	We agree this may be helpful for readers, so have now added it to the methods.	These databases were chosen following librarian advise and after an initial database scoping exercise on Google Scholar, as they seemed most suitable for balancing specify and comprehensively.	P4
Perhaps you might want to consider proving the full meaning	Thank you, this has been added	PerSPECTiF (Perspective, Setting, Phenomenon of interest, Environment	P4

of RETREAT and PerSPECTiF		(optional Comparison Timing, Findings) (21). RETREAT (Review question, Epistemology, Timescale, Resources, Expertise, Audience/purpose, Type of data) framework(17)	
In the data synthesis, it was unclear what data on study characteristics were extracted.	Thank you, we have added this detail. Also, based on your positive feedback, and having read the methods section again, we have moved details about data extraction which was placed under critical appraisal section to the data extraction section as we think it fits well there, and will readers appreciate how the data were extracted and synthesised. This also makes the critical appraisal a bit more consistent and easy to follow.		
While a through critique was provided for the evidence synthesis, not all the included studies had their risk of bias presented.	Thank you for identifying this was not clear. In the evidence synthesis table, it highlights bias noted by authors, this was not recorded by all, so not included for all. We have specified this in the supplementary file.	Limitations noted by authors (if any)	Supplementary material (p4)
In the section on 'accessibility defined through a candidacy lens', perhaps you might want to include study references.	This has been added	candidacy framework (13)	P5
The role of the community in shaping access to care is presented accordingly but it is interesting	Thank you, this is an interesting comment, this was not found to be a major theme in this review although has been noted in some literature and could warrant exploration. We have taken		

that there was no mention of the role of the media (e.g., television and radio) in influencing treatment access. The role of the media in accessing medical care for breast cancer was a finding in Agbeko et al. (2020).	note of your comment and will explore this further in our ongoing primary data collection aim to build on the findings of this review.		
Specific comment			
You may consider removing these two sentences as they seem superfluous because lines 35-36 have already highlighted how 'access' might obscure the dynamism between individuals, their health environment and healthcare professionals.	We agree and have deleted this sentence and agree it helps the paper read more concisely		P3
Perhaps you might want to be more specific about the number of qualitative studies.	We agree this would be insightful and so have clarified in the sentence	Twelve qualitative studies included some individual barriers to accepting care.	P6
You might want to consider rewriting this. For example, 'In ... studies (insert reference), this was not the	Thank you, we have rewritten to be clearer and added references	However, this was not always the primary focus but emerged in the findings(23,24).	P6

primary focus but emerged in the findings.			
You might want to consider re-writing OTC in full.	We have added the full word to ensure readers can follow	over the counter (OTC) medications	P7
Please correct the typo ' A lack of trusted...	We have amended this typo		P11
Reviewer 2			
Thanks for the opportunity to review this study. Neatly written paper. Some few concerns	Thank you for the encouraging feedback, we will address these concerns		
Abstract - Please be consistent with the use of the word Social-ecological or socio-ecological.	Thank you for spotting this typing error, we have amended to socio-ecological and checked the full text to ensure consistency		P2
The paper will benefit from proofreading to correct the grammatical and semiotical errors.	Thank you, the manuscript has now undergone a full proof read with all authors		
Titles for some of the figures were missing.	We have added the figure titles to the manuscript and figure images		P13 / figures

VERSION 2 – REVIEW

REVIEWER	Agyemang, Linda University of Southampton, Health Sciences
REVIEW RETURNED	12-Sep-2022
GENERAL COMMENTS	Thank you for revising the manuscript.